# National Early Warning Score 2 (NEWS2) as a prognostic tool for adult patients in emergency department: A retrospective observational study

Nor Safiahani Mhd Yunin¤◉, Toh Leong Tan◉¤◉*

Department of Emergency Medicine, Faculty of Medicine, Universiti Kebangsaan Malaysia.

◉ These authors contributed equally to this work.
¤ Current address: Department of Emergency Medicine, Faculty of Medicine, Universiti Kebangsaan Malaysia, Jalan Yaacob Latif, Bandar Tun Razak, 56000 Kuala Lumpur, Malaysia
* sebastianttl@yahoo.co.uk

## Abstract

The National Early Warning Score 2 (NEWS2) has been widely adopted to assess the severity of illness across various healthcare settings. However, its effectiveness as a prognostic tool specifically within the Emergency Department (ED) has not been validated in Malaysia. This retrospective observational study aimed to evaluate the performance of NEWS2 in predicting hospitalizations, critical care unit admissions, and in-hospital mortality among adult patients in a Malaysian ED, specifically at the Hospital Canselor Tuanku Muhriz. The discriminatory ability of NEWS2 was analysed using the area under the receiver operating characteristic curve. A total of 1906 adult patients were included in the study, with results indicating that NEWS2 demonstrated moderate to excellent performance in predicting outcomes. The AUROC values were 0.71 for hospitalization, 0.75 for critical care admission, and 0.86 for in-hospital mortality. Additionally, subgroup analyses for patients with sepsis and those with COVID-19 were also conducted. NEWS2 score however was not as effective in predicting critical outcomes for these two subgroups, limiting its utility as an early warning tool in diverse clinical scenarios. Overall, the results suggest that NEWS2 is a valuable tool for early risk stratification in the general adult population in the ED. Implementing NEWS2 into routine clinical practice could significantly enhance patient outcomes by promoting timely and informed decision-making, ultimately benefiting both patients and healthcare providers.

## Introduction

Overcrowding in the ED increases the risk of delayed treatment for critically ill patients [1]. There is a need for an effective, affordable, and easy-to-use prognostic tool to help mitigate this risk. Introduced by the Royal College of Physicians in 2012, the National Early Warning Score (NEWS) has emerged as a promising tool aimed

   

**Data availability statement:** All relevant data are within the manuscript and its Supporting Information files.

**Funding:** TLT received funding from Faculty of Medicine, Universiti Kebangsaan Malaysia (code FF-2023-135). The Funder had no role in the study design, data collection, analysis, decision to publish, or preparation of the manuscript.

**Competing interests:** NO authors have competing interests.

at enhancing the care of acutely ill patients [2]. The updated version, NEWS2, has demonstrated its potential across various healthcare settings due to its simplicity and effectiveness in evaluating illness severity and predicting patient outcomes. Despite its widespread implementation, the role of NEWS2 in the ED—specifically regarding its application to medical, trauma and surgical adult patients in Malaysia—has not been thoroughly investigated [2].

The healthcare burden of sepsis, related to its significant critical care unit admission and mortality rate calls for a simple yet effective one-for-all-tool to identify deteriorating patients early [3]. While sepsis has been studied extensively with various scoring systems namely SIRS, qSOFA, mSOFA and Shock index, the use of NEWS2 in predicting outcomes of ED patients in is yet to be validated.

In the recent COVID-19 pandemic, many EDs have become over-congested with more critically ill patients with high chance of deterioration. Overcrowding with limited resources necessitate an effective tool that can prioritize these critically ill patients appropriately. COVID-19 is a disease that primarily affects the respiratory system, with silent hypoxemia being its hallmark of clinical deterioration. Relative underscoring of hypoxemia in NEWS2 raises the concern of its ability to detect clinical deterioration early thus the delay in escalation of therapy for this subgroup [4,5].

Given its ability to predict patients' clinical deterioration objectively based on easily measured parameters, we considered the relevance of NEWS2 in our study. We hypothesized that a NEWS2 score of 5 and above will demonstrate strong predictive performance in identifying deteriorating diverse adult population early, our study aims to assess the performance of NEWS2 in prognosticating hospitalization, critical care unit admission, and in-hospital mortality rates among adult ED patients in a Malaysian context.

## Materials and methods

A single-centre retrospective observational study was conducted at the ED of Hospital Canselor Tuanku Muhriz (HCTM), Universiti Kebangsaan Malaysia. This study was approved by the Research Ethics Committee of HCTM (Code: UKM PPI/111/8/JEP-2023–275) and adhered to all ethical principles of Helsinki Declaration as well as confidentiality of patients' records. This is a retrospective study which informed consent was waived. The data collection period started from 19th February 2024 and end on 30th April 2024. All authors able to access the data during this period. Data between 1st April 2023 and 20th April were collected. Data were fully anonymized prior to collection, preventing authors from accessing any information that could identify individual participants during or after the data collection process.

All patients who presented to ED HCTM during the study period were considered for inclusion. Patients were excluded if they were less than 18 years old at the time of presentation, pregnant, brought in dead (BID), transferred from another healthcare facility with treatment already initiated, had an established do not resuscitate (DNR) status or those with missing vital signs documentation.

Data were extracted from patients' medical records both electronic and hardcopy, in addition to ED census data. Data that were collected include demographic information, comorbidities, vital signs, investigations, and patient outcomes (hospitalization,

critical care unit admission and in-hospital mortality). During the data collection, every piece of information gathered by the initial researcher was verified by another researcher. Following the cross-check, the two researchers get together to thoroughly review each patient's data for errors and ensure that all of the information is correct. We made every effort to ensure that the data collected was always of the highest calibre possible. We further performed subgroup analysis for sepsis and COVID-19. Sepsis is defined as having a Modified Sequential Organ Failure Assessment (mSOFA) score of 2 or more in patients suspected to have an infection [6,7]. The calculation of mSOFA score is based on 5 domains, each with a score between 0 and 4. NEWS2 score were then calculated.

NEWS2 is calculated based on 7 parameters, with each domain carrying a score between 0–3 [8]. The 7 parameters include heart rate (HR), respiration rate (RR), required supplemental oxygen, systolic blood pressure (sBP), temperature (T), oxygen saturation (SpO2) and level of consciousness. Oxygen saturation has 2 scales. Only one of the two scales should be used for each individual patient. Scale 2 is reserved for hypercapnic respiratory failure patients (e.g. COAD, obese patients, patients with chest wall deformities or neuromuscular disorders). Patients requiring supplemental oxygen to maintain satisfactory oxygen saturation will gain an extra 2 score on top of the calculated NEWS2. Maximum score is 20. The primary outcomes were hospital admission, critical care unit admission and in-hospital mortality.

### Statistical analyses

Statistical analyses were performed using SPSS software version 33. Descriptive statistics were used to summarize patient characteristics in frequency, mean or median accordingly. The Chi Square or Fisher's test was used for comparison of categorical data and independent T-test or Mann Whitney-U Test for continuous data depending on the normality distribution. P-value of less than 0.05 for a two-sided test was considered statistically significant. The Area Under the Receiver Operating Characteristic (AUROC) curve was applied to assess the discriminatory ability of NEWS2 provide in SPSS software. Sensitivity, specificity, positive predictive value (PPV), and negative predictive value (NPV), positive likelihood ratio (PLR), negative likelihood ratio (NLR) and accuracy were calculated for a cut-off point of 5 using online sensitivity and specificity calculation from MedCalc Easy-to-use statistical software

The sample size was calculated based on a study done by Thoren et. al., where sensitivity and specificity of NEWS2 for critical care unit admission and mortality prediction were 91.8% and 15.8% respectively for the general adult population [9]. With a prevalence of 85% of adult patients in ED HCTM (P = 0.85), z = 1.96 (CI = 95%) and w = 0.05, sample sizes for sensitivity (Sn) and specificity (Sp) are calculated as 136 and 1362 respectively. A NEWS2 cut of point of 5 or more was used for all three different outcomes being analysed. This is in conjunction with the Royal College of Physician 2017's recommendation for alert emergency response teams in hospitals to be activated when a NEWS2 threshold of 5 or more has been reached. Additional sample size for sepsis and COVID-19 subgroups were calculated separately. The sample size calculations for each sepsis and COVID-19 subgroup are detailed in the Supporting Information file 1. Using the biggest sample size calculated across all groups (n = 1589) and consideration of 20% missing data, this calculation determined that a total of 1906 samples were required.

## Results

A total of 2832 patients presented to the emergency department between 1st April 2023 and 20th April 2023. Of this, 926 patients were excluded from the study due to various factors including age less than 18 years old (n = 452), being pregnant (n = 47), established DNR status (n = 105), brought in death (n = 68), transferred from another healthcare facility (n = 27) and missing vital sign documentation (n = 227). Fig 1 illustrates this selection flowchart of patients included in the study.

### Patient clinical characteristics and demographic

1906 adult patients were analysed in this study, of which 52.6% were male, and 47.4% were female. Table 1 shows the demographics and clinical characteristics of patients that were included in the study.

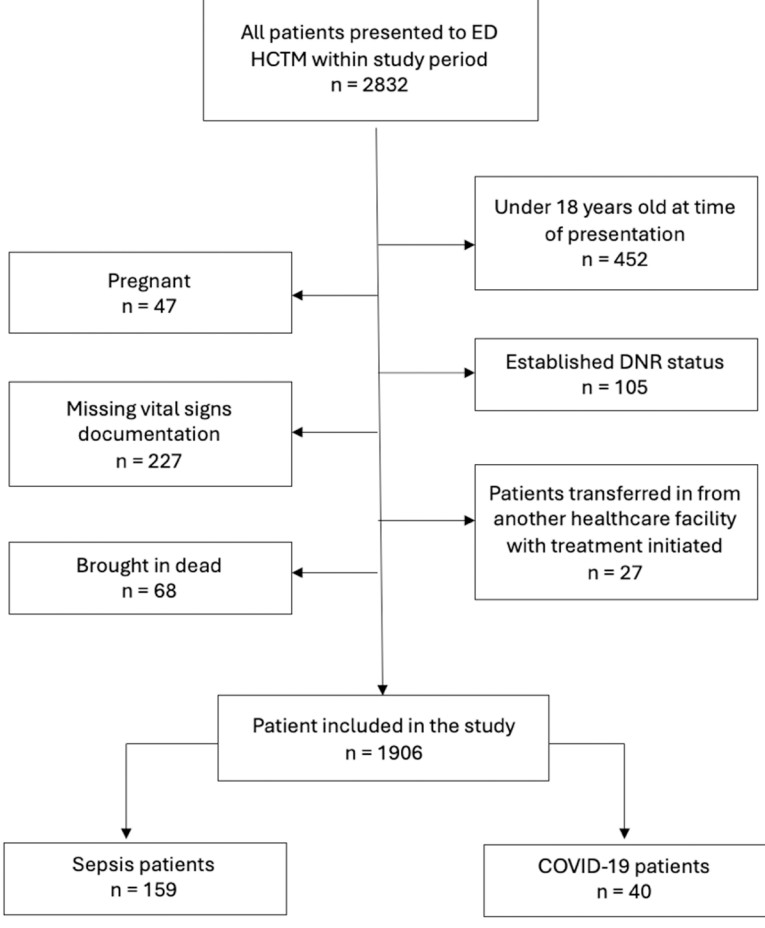

**Fig 1. The patients' selection flowchart.**

We further performed subgroups analyses for COVID-19 and sepsis patient. Out of 1906 patients that were included in the study, only 159 patients fulfilled the sepsis criteria, and only 40 patients were confirmed cases of COVID-19. The median NEWS2 score for sepsis and COVID-19 patients were 7 (IQR: 4–9) and 3 (IQR: 1–5) respectively.

## Outcomes of adult patients in ED

The outcomes of adult patients in the ED were assessed in terms of hospital admission, critical care unit admission, and mortality. Baseline characteristics such as gender, age, vital signs, comorbidities, diagnosis, and the NEWS2 are presented below in Table 2. Among the patients, 645 (33.8%) required hospitalization, 64 (3.3%) were admitted to the critical care unit, and 42 died due to various diagnosis, giving an overall mortality rate of 2.2%.

Out of 1906 patients, 645 (33.8%) were admitted to the hospital. There was no significant difference in hospital admission based on gender ($\chi^2 = 0.328$, $p = 0.328$). However, the median age of admitted patients (65 years, IQR: 47–74) was significantly higher compared to non-admitted patients (45 years, IQR 30–66, $p < 0.001$). Several baseline characteristics, such as HR, RR, dBP, T, and SpO2 showed statistically significant differences between the admitted and non-admitted groups (all $p < 0.05$). Additionally, patients with comorbidities like hypertension (60.2% vs. 31.3%), diabetes (42.3% vs. 20.7%), CVD (29.1% vs. 10.2%), chronic renal disease (16.0% vs. 5.5%), cerebrovascular accident (CVA) (8.2% vs. 2.5%), and malignancies (11.0% vs. 4.4%) had significantly higher rates of hospital admission (all $p < 0.05$).

**Table 1. Demographics and Clinical Characteristics of Patients Presented in ED.**

| Characteristics | All patients (n = 1906) |
|---|---|
| **Gender,** n (10) | |
| Male | 1002 (52.6) |
| Female | 904 (47.4) |
| **Age,** median (IQR) | 54 (34,70) |
| **Comorbidities,** n (10) | |
| Hypertension | 783 (41.1) |
| Diabetes | 534 (28.0) |
| Cardiovascular Diseases | 317 (16.6) |
| Chronic Obstructive Airway Diseases | 36 (1.9) |
| Chronic Renal Diseases | 172 (9.0) |
| Cerebral Vascular Accident | 84 (4.4) |
| Malignancies | 127 (6.7) |
| **Vital Signs,** median (IQR) | |
| Heart Rate, beats per min | 87 (76,100) |
| Respiratory Rate, breath per min | 18 (18,20) |
| Systolic Blood Pressure, mmHg | 134 (120,150) |
| Diastolic Blood Pressure, mmHg | 78 (69,87) |
| Temperature, °C | 37.0 (36.6,37.4) |
| Oxygen Saturation (SpO2), % | 98 (97,99) |
| **Required Supplemental Oxygen,** n (10) | 152 (8) |
| **State of Consciousness,** n (10) | |
| Alert | 1834 (96.2) |
| Altered consciousness | 72 (3.8) |
| **Diagnosis,** n (10) | |
| Medical illness | 729 (38.2) |
| Surgical disease | 438 (22.9) |
| Infectious illness | 613 (32.1) |
| Traumatic illness | 326 (17.1) |
| **Sepsis,** n (10) | 159 (8.3) |
| **COVID-19,** n (10) | 40 (2.1) |
| **NEWS2 score,** median (IQR) | |
| **Overall** | 1 (2,0) |
| **Sepsis** | 7 (4,9) |
| **COVID-19** | 3 (1,5) |

A total of 64 (3.3%) were admitted to a critical care unit. Although gender was not significantly associated with critical care unit admission ($\chi^2 = 0.061$, $p = 0.061$), patients admitted to the critical care unit were older, with a median age of 69 years (IQR 58–74), compared to those who were not (median age 53 years, IQR 33–69, $p < 0.001$). Significant differences were also noted in HR, RR, sBP, dBP, and SpO2 between the critical care and non-critical care groups (all $p < 0.05$). Comorbidities such as hypertension, diabetes, and CVD were significantly more common among patients admitted to critical care (all $p < 0.05$).

A total of 42 patients (2.2%) died during their stay in the ED. While gender differences were not statistically significant ($\chi^2 = 0.124$, $p = 0.124$), patients who died were significantly older (median age 69 years, IQR 54–74) compared to survivors (median age 54 years, IQR 33–69, $p < 0.001$). Vital signs, including HR, RR, sBP, dBP, T, and SpO2, were significantly different between survivors and non-survivors (all $p < 0.05$). Patients with comorbidities such as hypertension (69.0% vs. 40.5%), diabetes (47.6% vs. 27.6%), CVD (35.7% vs. 16.2%), and chronic renal disease (23.8% vs. 8.7%) had significantly higher mortality rates (all $p < 0.05$).

**Table 2. Characteristics of all adult patients in ED stratified by hospital admission, critical care unit admission and mortality.**

| Baseline Characteristics | Patient Outcome | | | | | | | | |
|---|---|---|---|---|---|---|---|---|---|
| | Hospital admission | | | Critical care unit admission | | | Mortality | | |
| | No | Yes | P value | No | Yes | P value | No | Yes | P value |
| **Total Patients, n (10)** | 1261 (66.2) | 645 (33.8) | | 1842 (96.6) | 64 (3.3) | | 1864 (97.8) | 42 (2.2) | |
| **Gender, n (10)** | | | | | | | | | |
| Male | 673 (53.4) | 329 (51.0) | 0.328a | 961 (52.2) | 41 (64.1) | 0.061a | 975 (52.3) | 27 (64.3) | 0.124a |
| Female | 588 (46.6) | 316 (49.0) | | 881 (47.8) | 23 (35.9) | | 889 (47.7) | 15 (35.7) | |
| **Age, median (IQR)** | 45 (30,66) | 65 (74,47) | <0.001b* | 53 (69,33) | 69 (74,58) | <0.001b* | 54 (69,33) | 69 (74,54) | <0.001b* |
| **Vital signs, median (IQR)** | | | | | | | | | |
| HR, beats per min | 85 (75,97) | 92 (79,107) | <0.001b* | 87 (100,76) | 88 (78,115) | 0.062b | 87 (76,99) | 112 (86,125) | <0.001b* |
| RR, breath per min | 18 (18,20) | 20 (18,22) | <0.001b* | 18 (20,18) | 20 (18,26) | <0.001b* | 18 (18,20) | 22 (20,26) | <0.001b* |
| sBP, mmHg | 134 (121,149) | 133 (118,153) | 0.503b | 134 (150,121) | 127 (104,156) | 0.042b* | 134 (121,150) | 120 (102,145) | 0.001b* |
| dBP, mmHg | 78 (70,86) | 77 (66,89) | 0.035b* | 78 (87,69) | 70 (51,89) | 0.014b* | 78 (69,87) | 69 (81,51) | 0.002b* |
| T, °C | 36.9 (36.6,37.2) | 37.0 (36.7,37.7) | <0.001b* | 37.0(37.4,36.6) | 36.9(36.5,37.4) | 0.684b | 36.9(36.6,37.4) | 37.4(38.2,36.7) | 0.001b* |
| SpO2, % | 98 (97,99) | 98 (95,99) | <0.001b* | 98 (99,97) | 96 (93,99) | <0.001b* | 98 (97,99) | 95 (98,91) | <0.001b* |
| **Required Supplemental oxygen, n (10)** | 4 (0.3) | 148 (22.9) | <0.001c* | 126 (6.8) | 26 (40.6) | <0.001a* | 129 (6.9) | 23 (54.7) | <0.001a* |
| **State of consciousness n(10)** | | | | | | | | | |
| Alert | 1247 (98.8) | 587 (91.0) | <0.001a* | 1784 (96.8) | 50 (78.1) | <0.001a* | 1804 (96.7) | 30 (71.4) | <0.001a** |
| **Comorbidities, n (10)** | | | | | | | | | |
| Hypertension | 395 (31.3) | 388 (60.2) | <0.001a* | 738 (40.1) | 45 (70.3) | <0.001a* | 754 (40.5) | 29 (69.0) | <0.001a* |
| Diabetes | 261 (20.7) | 273 (42.3) | <0.001a* | 508 (27.6) | 26 (40.6) | 0.022a* | 514 (27.6) | 20 (47.6) | 0.004a* |
| CVD | 129 (10.2) | 188 (29.1) | <0.001a* | 288 (15.6) | 29 (45.3) | <0.001a* | 302 (16.2) | 15 (35.7) | <0.001a* |
| COAD | 15 (1.2) | 21 (3.3) | 0.002a* | 33 (1.8) | 3 (4.7) | 0.118c | 33 (1.8) | 3 (7.3) | 0.043c* |
| Chronic Renal Disease | 69 (5.5) | 103 (16.0) | <0.001a* | 157 (8.5) | 15 (23.4) | <0.001a* | 162 (8.7) | 10 (23.8) | <0.001a* |
| CVA | 31 (2.5) | 53 (8.2) | <0.001a* | 80 (4.3) | 4 (6.3) | 0.526c | 80 (4.3) | 4 (9.5) | 0.110c |
| Malignancy | 56 (4.4) | 71 (11.0) | <0.001a* | 124 (6.7) | 3 (4.7) | 0.797c | 117 (6.3) | 10 (23.8) | <0.001a* |
| **Diagnosis n (10)** | | | | | | | | | |
| Medical illness | 395 (31.3) | 335 (51.9) | <0.001a* | 683 (37.0) | 47 (73.4) | <0.001a* | 714 (38.3) | 16 (38.0) | <0.001a* |
| Surgical disease | 312 (24.7) | 126 (19.5) | | 434 (23.5) | 4 (6.2) | | 430 (23.0) | 8 (19.0) | |
| Infectious illness | 313 (24.8) | 301 (46.6) | | 592 (32.1) | 22 (34.3) | | 584 (31.3) | 30 (71.4) | |
| Traumatic illness | 278 (22.0) | 48 (7.4) | | 325 (17.6) | 1 (1.5) | | 325 (17.4) | 1 (2.3) | |
| **NEWS2 median (IQR)** | 1 (0,1) | 2 (0,5) | <0.001b* | 1 (0,2) | 4 (1,9) | <0.001b* | 1 (0,2) | 7 (3,9) | <0.001b* |

*HR* heart rate, *RR* respiratory rate, *sBP* systolic blood pressure, *dBP* diastolic blood pressure; *T* temperature; *SpO2* oxygen saturation; *CVD* Cardiovascular disease; *COAD* Chronic Obstructive Airway Diseases; *CVA* cerebral vascular accident.

*A 2-sided *P* value of < 0.05 indicate statistical significance.

a Pearson Chi-square test.

b Mann-Whitney U test.

c Fisher's exact test.

The median NEWS2 score increased with the severity of the outcome: 1 (IQR: 0–2) for all patients, 2 (IQR: 0–5) for patients admitted to the hospital, 4 (IQR: 1–9) for patients admitted to critical care units, and 7 (IQR: 3–9) for those who died. A significant difference was observed in the median NEWS2 scores across these groups (p < 0.001). This analysis is presented in Fig 2.

Overall, we concluded that older age, abnormal vital signs, presence of comorbidities, altered consciousness and higher NEWS2 scores were associated with increased rates of hospital admission, critical care unit admission, and mortality.

**Performance of NEWS2 score in predicting outcomes for adult ED patients**

The performance of the NEWS2 score with cut-off of ≥5 in predicting hospitalization, critical care unit admission, and mortality of adult patients in ED was evaluated using the area under the receiver operating characteristic (AUROC) curve. Table 3 summarizes the ability of NEWS2 score to predict outcomes of adult ED patients.
The AUROC for NEWS2 in predicting hospitalization was 0.71 (95% CI: 0.68–0.73), indicating a moderate level of discrimination. This is shown in Fig 3.

NEWS2 score had an AUROC of 0.75 (95% CI: 0.67–0.82) for predicting critical care unit admission, indicating good discrimination (Fig 4).

For predicting mortality, the AUROC of the NEWS2 score was 0.86 (95% CI: 0.79–0.92), reflecting excellent discriminatory ability. This is shown in Fig 5. NEWS2 NPV was 99.1% (95% CI: 98.7–99.4) which was high.

The AUROC for NEWS2 in predicting mortality was 0.86 (0.79–0.92)

**Subgroup analysis: Sepsis patients in ED**

Out of 1906 patient, 159 patients fulfilled the sepsis criteria: presence of suspected infection and a Modified Sequential Organ Failure Assessment (mSOFA) score of 2 or more. Hospital admissions occurred in 98.7% of cases, critical care unit admissions in 13.2%, and mortality in 16.3%. In general, sepsis patients were noted to have higher NEWS2 score for all outcomes measured as compared to the general population. This analysis is shown in Fig 6.

The performance of NEWS2 (cutoff ≥ 5) was excellent in predicting sepsis hospital admission, with an AUROC of 0.97 (95% CI: 0.93–1.00). For critical care unit admissions, the AUROC was 0.75 (95% CI: 0.64–0.87), demonstrating moderate accuracy in predicting this outcome. However, the predictive capability for mortality was lower, with an AUROC of 0.66 (95% CI: 0.54–0.77). Baseline characteristics of sepsis patients in ED stratified by hospital admission, critical care unit admission and mortality is shown in S1 Table (Supporting information file 2)

**Subgroup analysis: COVID-19 patients in ED**

A total of 40 out of 1906 patients were diagnosis COVID-19 and included in the study. Hospitalization was required for 29 patients (72.5%), with 3 (7.5%) admitted to the critical care unit and 2 (5%) resulting in mortality. The NEWS2 score was significantly different among COVID-19 patients based on their outcomes, with highest scores associated with critical care admission (p < 0.001). This analysis is presented in Fig 7.

The performance of NEWS2 score ≥ 5 in predicting outcomes among COVID-19 patients varied across different outcomes in the Emergency Department. The results showed an AUROC of 0.68 (95% CI: 0.52–0.84) for hospital admission, 0.96 (95% CI: 0.90–1.02) for critical care unit admission and 0.59 (95% CI: 0.28–0.89) for in-hospital mortality. Baseline Characteristics of COVID-19 Patients in ED Stratified by Hospital Admission, Critical Care Unit Admission and Mortality is shown in S2 Table (Supporting Information file 2)

**Sensitivity analysis**

We further analysed the NEWS2 score with optimum cut-off level for each group. NEWS score showed moderate to excellent discriminatory ability in predicting outcomes among the overall adult patient, sepsis and COVID-19 in ED. The results

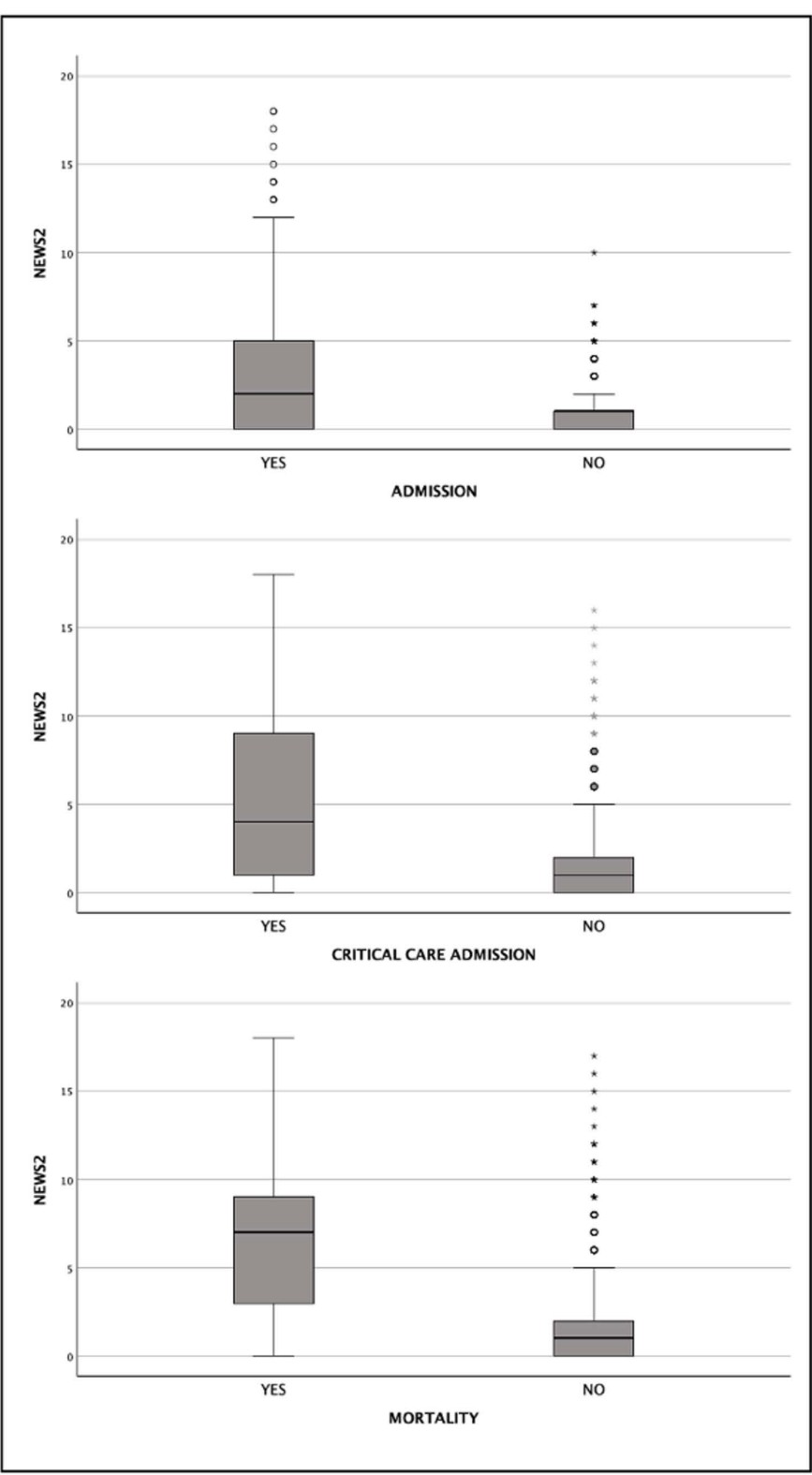

**Fig 2. Comparison of NEWS2 of adult patients in ED, based on outcome.**

**Table 3. The Performance of NEWS2 Score ≥5 in Predicting Outcomes of Adult ED Patients.**

| | Total Adult Patients n = 1906 | | |
|---|---|---|---|
| Outcome | Hospital Admission (n = 645) | Critical care unit admission (n = 64) | Mortality (n = 42) |
| AUROC | 0.71 (0.68-0.73) | 0.75 (0.67-0.82) | 0.86 (0.79-0.92) |
| Sn (10) | 29.4 (25.9-33.1) | 46.8 (34.2-59.7) | 66.7 (50.4-80.4) |
| Sp (10) | 98.1 (97.1-98.7) | 90.0 (88.5-91.3) | 90.0 (88.5-91.3) |
| PPV (10) | 88.7 (83.9-92.2) | 14.0 (10.8-17.9) | 13.0 (10.4-16.2) |
| NPV (10) | 73.1 (72.1-74.0) | 97.9 (97.4-98.4) | 99.1 (98.7-99.4) |
| PLR | 15.4 (10.2-23.4) | 4.6 (3.4-6.3) | 6.6 (5.1-8.6) |
| NLR | 0.72 (0.68-0.76) | 0.59 (0.47-0.74) | 0.37 (0.24-0.57) |
| P value | <0.001* | <0.001* | <0.001* |
| Accuracy | 74.8 (72.8-76.8) | 88.5 (87.0-89.9) | 89.5 (88.0-90.8) |

*A 2-sided P value of < 0.05 indicate statistical significance.

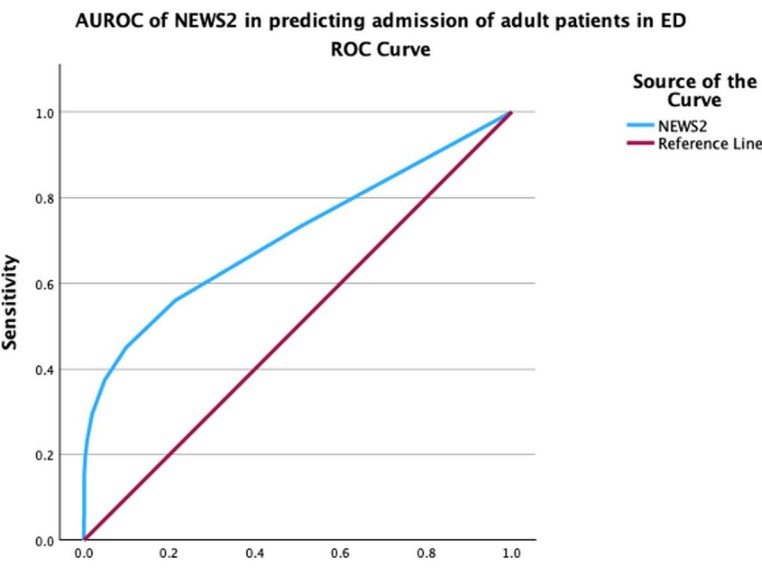

**Fig 3. AUROC of NEWS2 in predicting hospital admission of adult ED patients.**

aligned with the analyses that used a NEWS2 score threshold of 5. Table 4 summarizes the optimum NEWS2 score cut-off level and AUROC for all groups based on each outcome.

## Discussion

We examined the clinical profile of ED arrivals, specifically analysing the initial NEWS2 score and how it corresponded with three key outcomes: hospital admission, critical care admission, and in-hospital mortality. Our analysis demonstrated that the NEWS2 score showed excellent predictive capability across the general adult population. However, its predictive performance—as measured by AUROC—was not excellent when applied specifically to sepsis and COVID-19 patient subgroups.

Our research employed a NEWS2 score threshold of 5 or greater for all patient outcomes, in accordance with the Royal College of Physicians' recommendations, which designate this level as requiring an emergency response. Although

AUROC of NEWS2 in predicting critical care unit admission of adult patients in ED

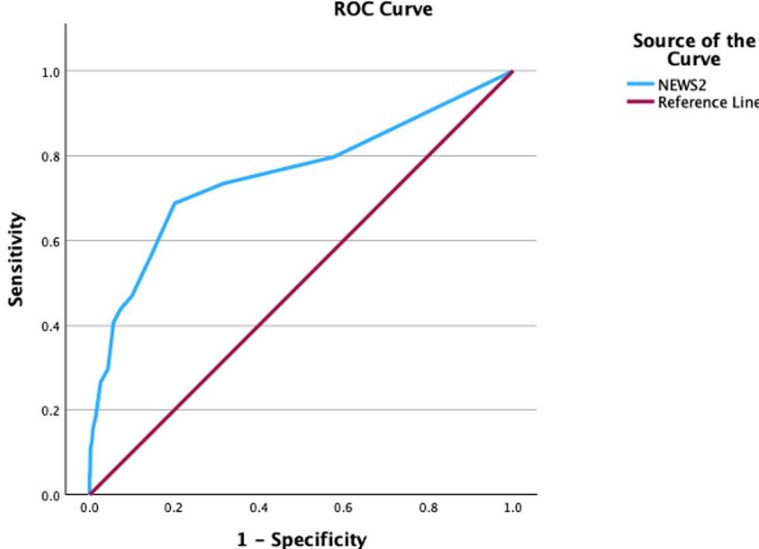

**Fig 4. AUROC of NEWS2 in predicting critical care unit admission of adult ED patients.**

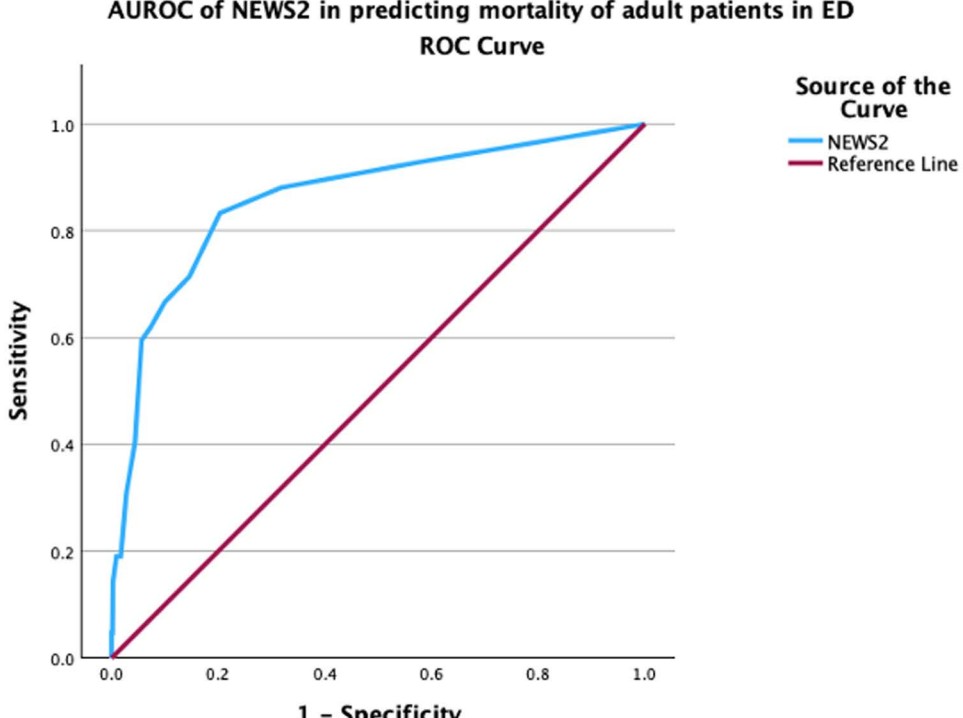

**Fig 5. AUROC of NEWS2 in predicting mortality of adult ED patients.**

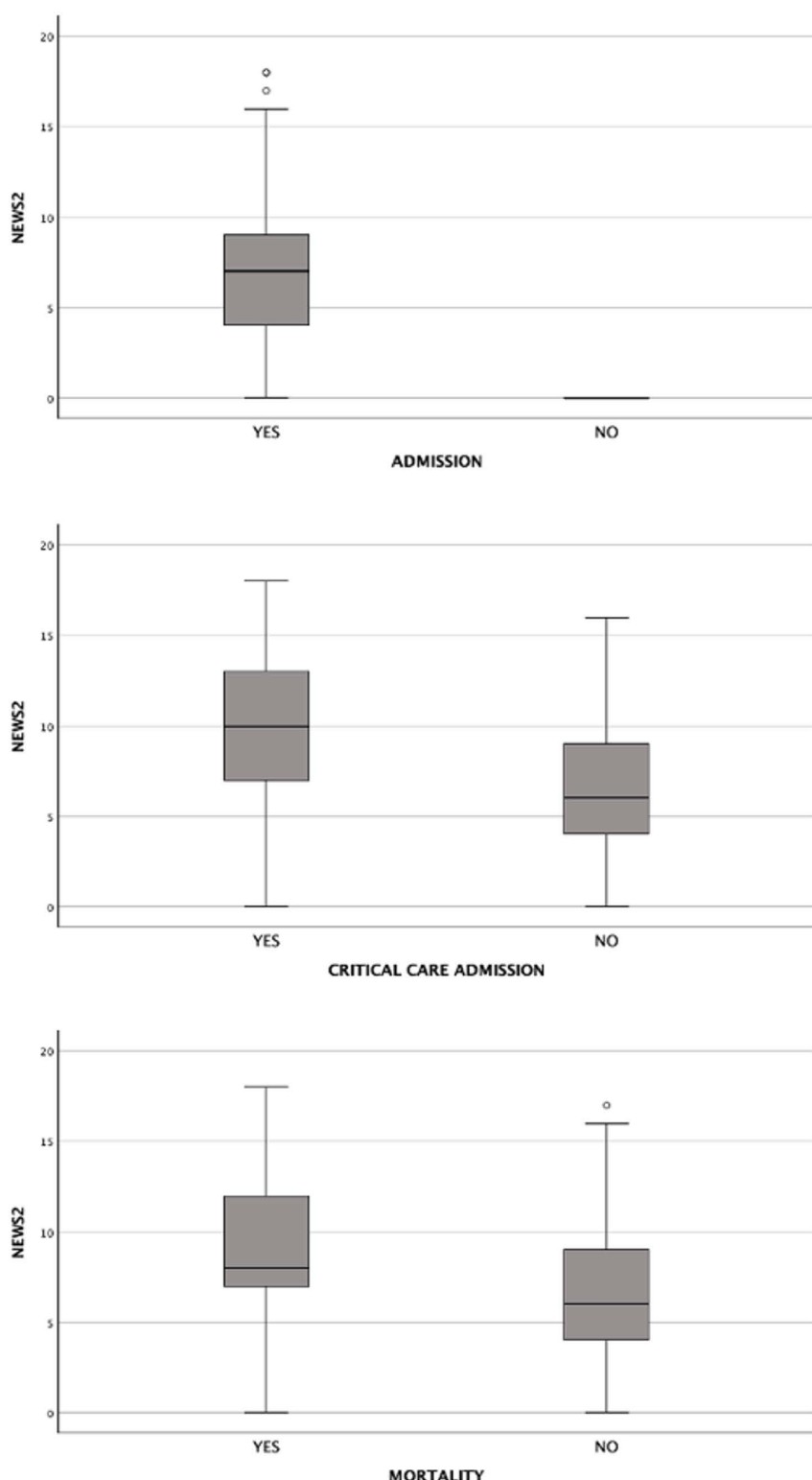

**Fig 6. Boxplot of NEWS2 scoring for sepsis patients in ED based on each outcome.**

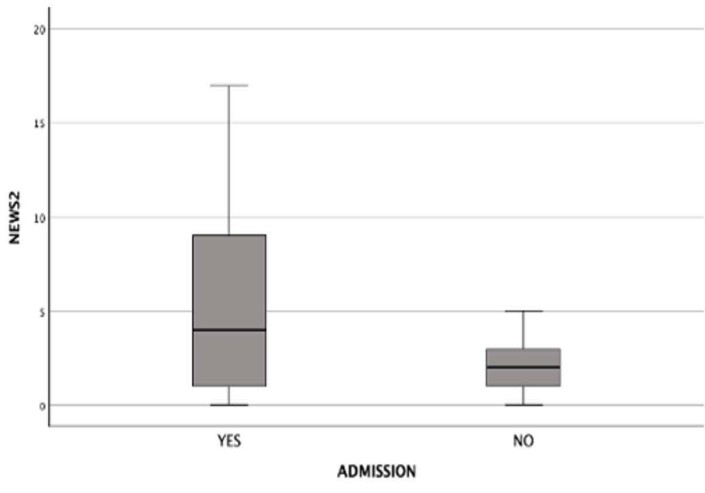

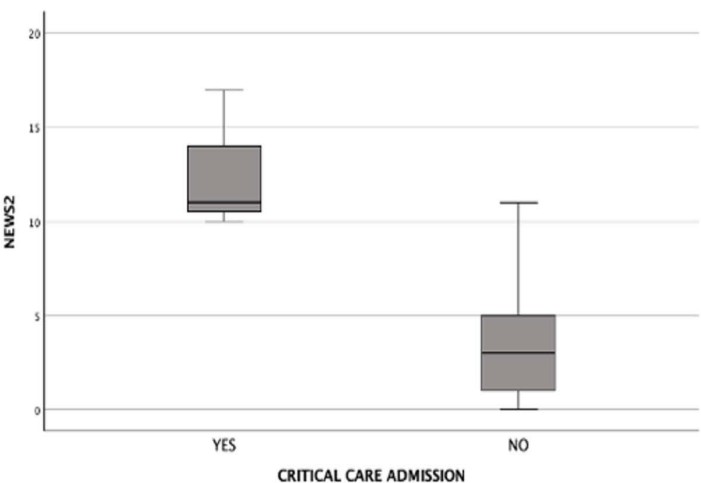

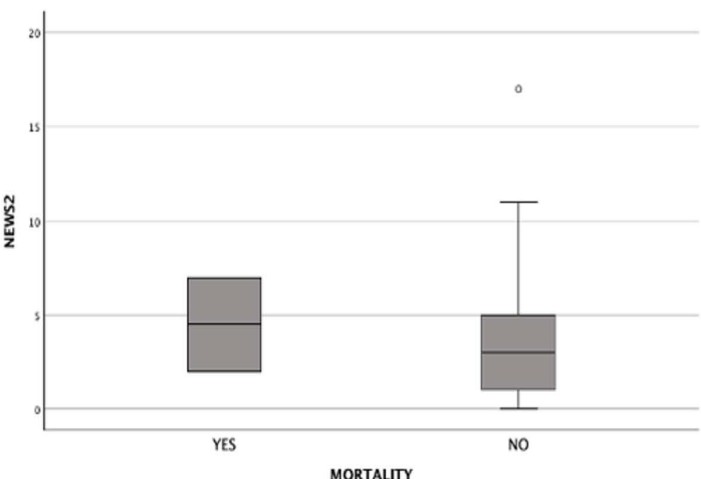

**Fig 7. Boxplot of NEWS2 scoring for COVID-19 patients in ED based on each outcome.**

**Table 4. Summary of optimum NEWS2 cut-off level and area under the receiver operating characteristic for all groups based on each outcome.**

| | Outcome | NEWS2 | AUROC |
|---|---|---|---|
| All patients | Hospital admission | 2 (0-5) | 0.71 (0.68-0.73) |
| | Critical care unit admission | 4 (12345678–9) | 0.75 (0.67-0.82) |
| | Mortality | 7 (345678–9) | 0.86 (0.79-0.92) |
| Sepsis patients | Hospital admission | 7 (45678–9) | 0.97 (0.93-1.00) |
| | Critical care unit admission | 10 (789101112–13) | 0.75 (0.64-0.87) |
| | Mortality | 8 (7891011–12) | 0.66 (0.54-0.77) |
| COVID-19 patients | Hospital admission | 4 (12345678–9) | 0.68 (0.52-0.84) |
| | Critical care unit admission | 11 (10111213141516–17) | 0.96 (0.90-1.02) |
| | Mortality | 4.5 (-,2) | 0.59 (0.28-0.89) |

customized thresholds for specific groups and outcomes might theoretically enhance predictive accuracy by improving sensitivity and specificity, practical implementation presents significant challenges. The operational complexity of managing variable thresholds across different patient populations and outcomes would likely prove impractical in busy clinical environments. Despite potential sacrifices in precision for certain patient subgroups, maintaining a standardized threshold—such as a NEWS2 score of 5—remains more operationally viable for broad clinical application.

Age, gender, and comorbidities such as hypertension, diabetes, and CVD significantly impacted patient outcomes, aligning with the findings of numerous studies conducted globally [10–12]. These patients' characteristics serve as fair predictors of adverse outcomes in acutely unwell patients presenting to the ED.

In our study, the rate of hospital admissions, critical care admission and in-hospital mortality lower compare with global ED data. NEWS2 demonstrated varying performance across different clinical outcomes, with the highest discriminatory ability in predicting mortality for all adult visit in ED. The sensitivity was relatively moderate for all outcomes, being highest for mortality, indicating that the test is not highly sensitive in detecting true positives. Specificity on the other hand was consistently high, ranging from 90.0% to 98.1%, which indicates a strong ability to correctly identify individuals who did not require admission or face mortality. Notably, the negative predictive values for critical care admission (97.9%) and mortality (99.1%) were very high, suggesting that the test is effective in ruling out these severe outcomes. Despite the moderate sensitivity, the high specificity and NPV make the test a valuable tool for identifying patients who are at low risk for critical outcomes, while the statistically significant P-values (<0.001) support these findings. For instance, a 2022 study from South Korea by Hong et al. reported hospital admission rates between 61.2% and 63.0%, but a lower mortality rate of 0.7%−0.8% [13]. In our centre, we have high volume of ED visit but lower admission rate. A more thorough investigation into the causes of this inconsistency will provide clearer understanding of the actual circumstances.

The performance of NEWS2 in this setting is consistent with previous studies that have validated the score as a reliable tool for risk stratification and clinical decision-making in the ED [11]. Furthermore, higher NEWS2 score was observed in patients experiencing severe outcomes, highlighting the need for vigilant monitoring and early intervention [2]. Trends of early warning score over time may be more helpful to understand the patient's condition and risk of deterioration rather than a single initial NEWS2 score calculated on arrival [14].

In our subgroup analyses, we found that the NEWS2 score fair in predicting critical outcomes for COVID-19 and sepsis patients, limiting its utility as an early warning tool in specific subgroup of patients. The inherent limitations of the NEWS2 score in capturing the complex pathophysiology of COVID-19 and the multifaceted factors contributing to mortality in these patients may explain the observed differences in its predictive ability [12,15,16]. These findings underscore the need for clinicians to exercise caution and nuance when interpreting and applying the NEWS2 score, particularly in the context of complex patient populations.

It is important to note that the performance of the NEWS2 score may be influenced by various factors, including the age [12], specific population characteristics [4], and healthcare setting [17]. Further research is warranted to explore the underlying reasons for the observed differences in predictive accuracy across different patient groups and to investigate ways to enhance the score's performance, particularly in the context of emerging infectious diseases like COVID-19.

The findings of our study reveal varying performance of the NEWS2 score as an outcome predictor among sepsis patients in the ED. This suggests that while the NEWS2 score is effective in the initial assessment of sepsis patients, it may not fully capture the complexities of disease progression or the need for critical care interventions. These findings align with previous studies that have highlighted the strengths of NEWS2 in predicting hospital and critical care admissions [8]. However, its reduced utility for mortality prediction underscores the need for combined or alternative metrics to enhance sepsis management in the ED.

The NEWS2 score was effective in predicting outcomes among COVID-19 patients, with higher scores indicating an increased risk of hospitalization, critical care admission and mortality [4]. In our study, higher NEWS2 score however did not correlate with a worse outcome in COVID-19. With regards to the limited sample size, although 1906 patients were included in the study, 856 were negative for COVID upon ED presentation and 1010 were not subjected for COVID-19 test, leaving only 40 confirmed cases of COVID-19. The small sample size limited the ability to draw a statistically significant conclusion for the prognostic ability of NEWS2 in predicting outcome of COVID-19 patients in ED.

When NEWS2 was initially developed, its use was limited to patients in medical wards, but over time many have incorporated its usability across the acute care pathway [18]. The strength of our study, we included all adult patients presenting to the ED regardless of the nature of the disease (i.e., medical, surgical, trauma and infectious disease) to maximise patient diversity. Reaching the targeted sample size indicates that the results of our study are robust.

## Limitation of the study

The retrospective, single-centre design of this study potentially limits the generalizability of its findings, as it may not fully capture the diversity of patient populations or variations in clinical practices across different healthcare settings. Since patient demographics and admission criteria can differ between hospitals, the external validity of these results requires further validation in emergency departments at various institutions.

Despite these constraints, this study offers valuable insights into the NEWS2 score's performance as a prognostic tool in the ED setting. It illuminates both the strengths and limitations of the NEWS2 score in predicting clinical outcomes for adult patients, including those with sepsis and COVID-19, thereby enhancing our understanding of its utility in acute care contexts. Our study found lower prevalence rates for sepsis (8.3%) and COVID-19 (2.1%) than expected based on our sample size calculations, which had anticipated sepsis prevalence of 25% [19] and COVID-19 prevalence of 5% (based on unpublished in-house pilot data). This discrepancy resulted in smaller subgroup sample sizes, limiting our ability to draw robust conclusions about the NEWS2 score's utility specifically for sepsis and COVID-19 patients in this critical care environment.

## Conclusion

In conclusion, NEWS2 score shows promise as an early warning system for predicting hospital admissions, critical care requirements, and mortality among diverse adult emergency department patients. However, its effectiveness varies across patient subgroups and outcomes, highlighting the need for continued assessment and refinement, especially for high-risk populations like those with sepsis or COVID-19.

Future research should focus on improving risk prediction models and developing targeted interventions for specific high-risk groups to enhance outcomes and reduce healthcare resource strain. Exploring the causes of gender disparities in critical care outcomes could inform gender-specific healthcare approaches.

Our study emphasizes the importance of early identification and management of high-risk ED patients, particularly older males with multiple comorbidities. NEWS2 score proves to be a reliable, simple, and effective tool for predicting mortality, hospital admission, and critical care needs. Using such a structured assessment tool can help clinicians evaluate illness severity and make timely management decisions, while underscoring the value of early risk stratification. Incorporating NEWS2 into everyday clinical practice may lead to improved patient outcomes.

## Supporting information

**S1 Table. Baseline characteristics of sepsis patients in ED stratified by hospital admission, critical care unit admission and mortality.**
(DOCX)

**S2 Table. Baseline characteristics of COVID-19 patients in ED stratified by hospital admission, critical care unit admission and mortality.**
(DOCX)

## Author contributions

**Conceptualization:** Nor Safiahani Mhd Yunin, Toh Leong Tan.

**Data curation:** Nor Safiahani Mhd Yunin.

**Formal analysis:** Nor Safiahani Mhd Yunin, Toh Leong Tan.

**Investigation:** Nor Safiahani Mhd Yunin, Toh Leong Tan.

**Methodology:** Nor Safiahani Mhd Yunin, Toh Leong Tan.

**Project administration:** Nor Safiahani Mhd Yunin, Toh Leong Tan.

**Supervision:** Toh Leong Tan.

**Visualization:** Nor Safiahani Mhd Yunin.

**Writing – original draft:** Nor Safiahani Mhd Yunin.

**Writing – review & editing:** Nor Safiahani Mhd Yunin, Toh Leong Tan.

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
