## [Decision Letter · Decision Letter 0]

6 May 2025

Dear Dr. Tan,

Thank you for submitting your manuscript to PLOS ONE. After careful consideration, we feel that it has merit but does not fully meet PLOS ONE’s publication criteria as it currently stands. Therefore, we invite you to submit a revised version of the manuscript that addresses the points raised during the review process.

We look forward to receiving your revised manuscript.

Kind regards,

Yaser Mohammed Al-Worafi

Academic Editor

PLOS ONE

 [TLT received funding from Faculty of Medicine, Universiti Kebangsaan Malaysia (code FF-2023-135)]. 

Reviewers' comments:

Reviewer's Responses to Questions

**Comments to the Author**

1. Is the manuscript technically sound, and do the data support the conclusions?

Reviewer #1: Yes

Reviewer #2: Yes

2. Has the statistical analysis been performed appropriately and rigorously?

Reviewer #1: Yes

Reviewer #2: Yes

3. Have the authors made all data underlying the findings in their manuscript fully available?

Reviewer #1: Yes

Reviewer #2: Yes

4. Is the manuscript presented in an intelligible fashion and written in standard English?

Reviewer #1: Yes

Reviewer #2: Yes

Reviewer #1: Thank you for inviting me to review the manuscript, ‘National Early Warning Score 2 (NEWS2) as a prognostic tool for adult patients in the emergency department: A retrospective observational study’. In this study, the authors have set out to validate the prognostic performance of the NEWS2 score in ED patients for admission to the hospital, admission to the critical care unit and in-hospital mortality. Overall, the manuscript is well written and the study is interesting. However, I wish to seek clarification, and comment on a few points. They are as follows:

1. The authors state that the ‘data were fully anonymised prior to collection’. Is this correct?

2. I presume the data collection was manual. If so, what measures were taken to mitigate transcription errors?

3. The authors state that the mSOFA score has 4 domains. Grissom’s mSOFA score actually has 5 domains. Please specify which domain was dropped?

4. The authors state that additional sample size calculations were made for sepsis and COVID-19. What were they?

5. Many of the data points in Table 1 are repeated in the narrative. This duplication should be removed.

6. The results section should carry factual information only. Any opinions and interpretations should be moved to the discussion.

Reviewer #2: This study makes significant advances to our understanding of the NEWS2 score's usefulness as an early warning system in emergency departments. Its results confirm that NEWS2 is useful for identifying patients who are at danger, but the authors ought to be more clear about the study's limitations. The NEWS2 score's performance varies among specific subgroups, such as patients with sepsis and those with COVID-19, and this has to be explained more clearly because it may have an impact on the score's therapeutic applicability and generalizability across a range of patient demographics.

Furthermore, the manuscript would benefit from an expanded analysis or discussion of gender disparities observed in patient outcomes. Exploring how gender influences the predictive accuracy or response to clinical deterioration could enhance the paper's depth and promote awareness of the need for personalized and equitable care strategies.

To further strengthen the manuscript, the authors are advised to provide a thorough and transparent account of the statistical methods used, including justification for their selection and application. This will ensure that readers and reviewers can confidently assess the validity and robustness of the study’s conclusions.

Importantly, no ethical concerns have been identified in relation to the conduct of the research, and there is no indication of issues with dual or redundant publication. These points, combined with the above suggestions, would help refine and enhance the manuscript’s clarity, scientific rigor, and practical relevance.

**Do you want your identity to be public for this peer review?** For information about this choice, including consent withdrawal, please see our Privacy Policy

Reviewer #1: No

Reviewer #2: **Yes: ** Emmanuel Kwasi Acheampong

---

## [Author Response · Author response to Decision Letter 1]

7 May 2025

We had already revised the manuscript accordingly. We had replied all the comments in the rebuttal letter. A revised manuscript and a manuscript with track changes were uploaded into the system. Thank you very much and we hope the revised manuscript will receive your kind consideration for publication.

Thank you very much for reviewing our manuscript. Below are our replies to the reviewers:

Academic editor comments:

Answer: The manuscript had been revised according to PLOS ONE's style requirements.

[TLT received funding from Faculty of Medicine, Universiti Kebangsaan Malaysia (code FF-2023-135)].

Please state what role the funders took in the study. If the funders had no role, please state: ""The funders had no role in study design, data collection and analysis, decision to publish, or preparation of the manuscript."If this statement is not correct you must amend it as needed.

Answer: We would like to revise the financial disclosure as below, Thank you:

[TLT received funding from Faculty of Medicine, Universiti Kebangsaan Malaysia (code FF-2023-135). The Funder had no role in the study design, data collection, analysis, decision to publish, or preparation of the manuscript.]

Answer: Supporting Information Captions were inserted at the end of manuscript and cited accordingly.

Answer: We had removed the following citation and updated the list of references.

“6. Physicians RCo. National Early Warning Score (NEWS) 2: Standardising the assessment of acute-illness severity in the NHS. London: RCP; 2017.” because it was retracted.

Reviewer #1 comments:

Thank you for inviting me to review the manuscript, ‘National Early Warning Score 2 (NEWS2) as a prognostic tool for adult patients in the emergency department: A retrospective observational study’. In this study, the authors have set out to validate the prognostic performance of the NEWS2 score in ED patients for admission to the hospital, admission to the critical care unit and in-hospital mortality. Overall, the manuscript is well written and the study is interesting. However, I wish to seek clarification, and comment on a few points. They are as follows:

1. The authors state that the ‘data were fully anonymised prior to collection’. Is this correct?

Answer: We gather information from the hospital's census list, which was initially anonymized. The census contains only the case registration number, not the patient's name.

2. I presume the data collection was manual. If so, what measures were taken to mitigate transcription errors?

Answer: Every piece of information gathered by the initial researcher was verified by another researcher. Following the cross-check, the two researchers get together to thoroughly review each patient's data for errors and ensure that all of the information is correct. We made every effort to ensure that the data collected was always of the highest calibre possible. We had included these statements in the manuscript as well.

3. The authors state that the mSOFA score has 4 domains. Grissom’s mSOFA score actually has 5 domains. Please specify which domain was dropped?

Answer: We appreciate you bringing this up. We made a typo. There were five domains. The correction had already been made. Five domains were used in this study.

4. The authors state that additional sample size calculations were made for sepsis and COVID-19. What were they?

Answer: Thank you for this comment. The sample size calculations for each sepsis and COVID-19 subgroup are detailed in the Supporting Information file. Sample sizes for the sepsis and COVID-19 subgroups are n=1589 and n=602, respectively.

We had included this phrase in the manuscript:

“The sample size calculations for each sepsis and COVID-19 subgroup are detailed in the Supporting Information file.”

5. Many of the data points in Table 1 are repeated in the narrative. This duplication should be removed.

Answer: Thank you for this comment. We had removed all the duplicated narrative.

6. The results section should carry factual information only. Any opinions and interpretations should be moved to the discussion.

Answer: Thank you for the positive comment. We had moved all the interpretation to the discussion segment.

Reviewer #2 comments:

This study makes significant advances to our understanding of the NEWS2 score's usefulness as an early warning system in emergency departments. Its results confirm that NEWS2 is useful for identifying patients who are at danger, but the authors ought to be more clear about the study's limitations.

1. The NEWS2 score's performance varies among specific subgroups, such as patients with sepsis and those with COVID-19, and this has to be explained more clearly because it may have an impact on the score's therapeutic applicability and generalizability across a range of patient demographics.

Answer: Thank you for this comment. The healthcare burden of sepsis, related to its significant critical care unit admission and mortality rate calls for a simple yet effective one-for-all-tool to identify deteriorating patients early (1). While sepsis has been studied extensively with various scoring systems namely SIRS, qSOFA, mSOFA and Shock index, the use of NEWS2 in predicting outcomes of ED patients in is yet to be validated.

In the recent COVID-19 pandemic, many EDs have become over-congested with more critically ill patients with high chance of deterioration. Overcrowding with limited resources necessitate an effective tool that can prioritize these critically ill patients appropriately. COVID-19 is a disease that primarily affects the respiratory system, with silent hypoxemia being its hallmark of clinical deterioration. Relative underscoring of hypoxemia in NEWS2 raises the concern of its ability to detect clinical deterioration early thus the delay in escalation of therapy for this subgroup (2, 3).

1. Tan, T. L., Ahmad, N. H. & Neoh, H. M. 2018. Sepsis: The New Insight. Malaysia: Penerbit Universiti Kebangsaan Malaysia.

2. Baker KF, Hanrath AT, van der Loeff IS, Kay LJ, Back J, Duncan CJ. National Early Warning Score 2 (NEWS2) to identify inpatient COVID-19

3. Myrstad, M., Ihle-Hansen, H., Tveita, A. A., Andersen, E. L., Nygard, S., Tveit, A. & Berge, T. 2020. National Early Warning Score 2 (News2) on Admission Predicts Severe Disease and in-Hospital Mortality from Covid-19 - a Prospective Cohort Study. Scand J Trauma Resusc Emerg Med 28(1): 66.

We had inserted these statements in introduction segment.

2. Furthermore, the manuscript would benefit from an expanded analysis or discussion of gender disparities observed in patient outcomes. Exploring how gender influences the predictive accuracy or response to clinical deterioration could enhance the paper's depth and promote awareness of the need for personalized and equitable care strategies.

Answer: We appreciate your thoughtful remark. Males were shown to have more critical care admissions and deaths than females in number, however the differences were not statistically significant. Additional analysis is deemed inappropriate. We regret this ambiguous information. We had removed the ambiguous statement in the discussion segment.

3. To further strengthen the manuscript, the authors are advised to provide a thorough and transparent account of the statistical methods used, including justification for their selection and application. This will ensure that readers and reviewers can confidently assess the validity and robustness of the study’s conclusions.

Answer: Thank you for the comment. We had improved the manuscript statistical analyses segment to make it more thorough to ensure the robustness.

The revised manuscript as below:

“Statistical analyses were performed using SPSS software version 33. Descriptive statistics were used to summarize patient characteristics in frequency, mean or median accordingly. The Chi Square or Fisher’s test was used for comparison of categorical data and independent T-test or Mann-Whitney U Test for continuous data depending on the normality distribution. P-value of less than 0.05 for a two-sided test was considered statistically significant. The Area Under the Receiver Operating Characteristic (AUROC) curve was applied to assess the discriminatory ability of NEWS2 provide in SPSS software. Sensitivity, specificity, positive predictive value (PPV), and negative predictive value (NPV), positive likelihood ratio (PLR), negative likelihood ratio (NLR) and accuracy were calculated for a cut-off point of 5 using online sensitivity and specificity calculation from MedCalc Easy-to-use statistical software”.

4. Importantly, no ethical concerns have been identified in relation to the conduct of the research, and there is no indication of issues with dual or redundant publication. These points, combined with the above suggestions, would help refine and enhance the manuscript’s clarity, scientific rigor, and practical relevance.

Answer: Thank you for the wonderful complement, and we truly value your comment.

---

## [Editor Report · Decision Letter 1]

22 May 2025

National Early Warning Score 2 (NEWS2) as a prognostic tool for adult patients in emergency department: A retrospective observational study.

PONE-D-25-11794R1

Dear Dr. Tan,

We’re pleased to inform you that your manuscript has been judged scientifically suitable for publication and will be formally accepted for publication once it meets all outstanding technical requirements.

Kind regards,

Yaser Mohammed Al-Worafi

Academic Editor

PLOS ONE
---

## [Editor Report · Acceptance letter]

PONE-D-25-11794R1

PLOS ONE

Dear Dr. Tan,

I'm pleased to inform you that your manuscript has been deemed suitable for publication in PLOS ONE. Congratulations! Your manuscript is now being handed over to our production team.

Kind regards,

on behalf of

Professor Yaser Mohammed Al-Worafi

Academic Editor

PLOS ONE